The characteristics of host lipid body biogenesis during coral-dinoflagellate endosymbiosis

Chen Hung-Kai 1
Rosset Sabrina L. 1
Wang Li-Hsueh 1 2
Chen Chii-Shiarng cchen@nmmba.gov.tw 1 2 3
1 National Museum of Marine Biology and Aquarium , Pingtung , Taiwan
2 Graduate Institute of Marine Biology, National Dong-Hwa University , Pingtung , Taiwan
3 Department of Marine Biotechnology and Resources, National Sun Yat-sen University , Kaohsiung , Taiwan
Levy Oren
Electronic publication date: 2021 Jun 23
Publication date: 2021
Volume: 9
Electronic Location ID: e11652
Received 2021 Feb 26; Accepted 2021 May 31
Copyright: ©2021 Chen et al.
Copyright year: 2021
Copyright holder: Chen et al.
License: This is an open access article distributed under the terms of the Creative Commons Attribution License, which permits unrestricted use, distribution, reproduction and adaptation in any medium and for any purpose provided that it is properly attributed. For attribution, the original author(s), title, publication source (PeerJ) and either DOI or URL of the article must be cited.
License URL: https://creativecommons.org/licenses/by/4.0/

Keywords: Coral, Diel cycle, Endoplasmic reticulum, Lipid bodies, Symbiosis

Funding: Ministry of Science and Technology (MOST) of Taiwan NSC 101-2311-B-291-002-MY3 MOST 108-2311-B-291-001 NMMBA 99200311 This work was supported by a grant from the Ministry of Science and Technology (MOST) of Taiwan (formerly the National Science Council [NSC]; NSC 101-2311-B-291-002-MY3 and MOST 108-2311-B-291-001 to CSC) and by intramural funding from NMMBA (99200311). The funders had no role in study design, data collection and analysis, decision to publish, or preparation of the manuscript.

==============================
Intracellular lipid body (LB) biogenesis depends on the symbiosis between coral hosts and their Symbiodinaceae. Therefore, understanding the mechanism(s) behind LB biosynthesis in corals can portentially elucide the drivers of cellular regulation during endosymbiosis. This study assessed LB formation in the gastrodermal tissue layer of the hermatypic coral Euphyllia glabrescens. Diel rhythmicity in LB size and distribution was observed; solar irradiation onset at sunrise initiated an increase in LB formation, which continued throughout the day and peaked after sunset at 18:00. The LBs migrated from the area near the mesoglea to the gastrodermal cell border near the coelenteron. Micro-LB biogenesis occurred in the endoplasmic reticulum (ER) of the host gastrodermal cells. A transcriptomic analysis of genes related to lipogenesis indicated that binding immunoglobulin protein (BiP) plays a key role in metabolic signaling pathways. The diel rhythmicity of LB biogenesis was correlated with ER-localized BiP expression. BiP expression peaked during the period with the largest increase in LB formation, thereby indicating that the chaperoning reaction of abnormal protein folding inside the host ER is likely involved in LB biosynthesis. These findings suggest that the host ER, central to LB formation, potentially facilitates the regulation of endosymbiosis between coral hosts and Symbiodiniaceae.

Introduction

Endosymbiosis between coral hosts and dinoflagellates of the Symbiodiniaceae family is of global importance because it serves as the energenic foundation for all coral reefs. However, systematic understanding of how this association functions at the cellular and molecular levels is lacking. A hallmark of symbiosis is the daily rhythmic lipid body (LB) formation and subsequent degradation within the gastrodermal cells of coral hosts (Chen et al., 2012). Metabolic exchange is the basis of coral–algal endosymbiosis, and we previously proposed that these LBs function as centers of this metabolic dialog (Chen et al., 2017). Therefore, LB formation is a compelling target for further research aimed at eucidating mechanisms of endosymbiosis.

LBs are ubiquitous organelles, thought to originate in specialized compartments of the smooth endoplasmic reticulum (ER) (as shown in eukaryotes, see Wilfling et al., 2014) and later bud off into the cytosol. This hypothesis is based on the location of key enzymes involved in storage lipid biosynthesis in the ER and the discovery of ER-specific proteins in LBs (Jacquier et al., 2011). According to the model proposed by Wilfling et al. (2014), storage lipids accumulate between sheaths of the ER membrane, and the recruitment of enzymes (involved in lipid metabolism) and accessory proteins (involved in LB budding and maturation) to the LB formation site results in LB growth. Whether mature LBs completely separate from or remain connected to the ER may vary across biological systems.

Binding immunoglobulin protein (BiP), also known as glucose-regulated protein 78 (GRP78), is an ER-specific protein and a member of the heat shock 70 protein family. Moreover, it is a central regulator of unfolded protein response (UPR) activation to ER stress (Bertolotti et al., 2000) and is associated with purified coral LBs and other lipid droplets (Peng et al., 2011; Prattes et al., 2000; Zhang et al., 2016). Furthermore, by dynamically interacting with organelles such as the Golgi apparatus, mitochondria, endosomes, lysosomes, and lipid droplets through membrane contact sites (Cohen, Valm & Lippincott-Schwartz, 2018; Wu, Carvalho & Voeltz, 2018), the ER is generally thought to be involved in not only conveying crucial cellular signals but also lipid metabolism (Han & Kaufman, 2016).

Light irradiation drives daily LB formation in corals, suggesting that photosynthetically fixed carbon is translocated to coral hosts and incorporated into the following storage lipids, constituting the cores of the LBs: triacylglycerols, sterol esters, and wax esters (Chen et al., 2012). Analyses of the lipidomic profiles of dinoflagellates, LBs, and host gastrodermal cells have suggested that hosts and endosymbionts both contribute fatty acid (FA) moieties to LB biosynthesis (Chen et al., 2017). Furthermore, a study (Chen et al., 2012) revealed that LB density peaks at sunset and returns to baseline overnight; the study also demonstrated that LBs migrate across the host gastroderm during the diel cycle, and the highest densities were observed near the mesoglea at noon and near the coelenteron at night. However, the in situ tissue architecture of coral tentacles used in the study was highly distorted because amputated tentacles were contracted.

Here, we tested the hypothesis that LB biosynthesis in corals occurs in the host ER. We reconfirmed this LB migration trajectory by using an improved protocol, and characterized the process of coral LB maturation using transmission electron microscopy (TEM) to analyze diel fluctuations in LB size and distribution. In addition, transcriptomic data related to UPR activation were used to elucidate the correlation among BiP, ER, and lipogenesis throughout the diel cycle. Complemetary to the transcriptomic-level analysis, LB biosynthesis in corals was found to occur in the host ER when the abundance of the ER marker protein BiP in isolated LBs was measured. Understanding intracellular LB formation and maturation processes in corals can help to elucidate the mechanisms behind coral–algae endosymbiosis; such information could make tangible contributions to coral lipidomics in general, and more specifically, can be applied to research geared toward understanding and mitigating coral bleaching (i.e., the breakdown of symbiosis).

Materials & Methods

Coral husbandry

Corals (Euphyllia glabrescens) were collected from a reef at the inlet of the Third Nuclear Power Plant in Nanwan Bay, Taiwan (21∘57.376′N, 120∘ 45.291′E) in 2012. Coral collection was approved by the Kenting National Park Management Office. Ten∼ twelve colonies were cultured over eight years in each 4-ton tanks with flow-through seawater (exchange rate of ∼80L/h) in the husbandry center of the National Museum of Marine Biology and Aquarium (NMMBA). Twenty-four colonies for experiments were maintained under an ambient diel cycle with natural sunlight, and temperature was maintained at 26.5 ±1 ∘C using a microprocessor-controlled cooler (First FC-45, Aquatech, Kaohsiung, Taiwan). Light intensity and water temperature were continuously recorded using a HOBO Pendant® logger placed near the colonies in each tank (UA-002; Onset, Pocasset, MA, USA).

Analysis of in situ LB distribution

Tentacle samples were collected at the following nine times during one diel cycle: 06:00 (sunrise), 9:00, 12:00 (noon), 15:00, 18:30 (sunset), 21:00, 00:00 (midnight), 03:00, and 05:00. Tentacles were collected from three distinct colonies (n = 3 tentacles/colony) at each time point. A single polyp was targeted only once to avoid sampling-induced stress. For accurate analysis of LBs, preserving the natural tissue architecture of the expanded tentacle was critical. Sampling the tentacle without using a capillary clip resulted in a sectioned gastroderm exhibiting multiple layers of cells containing endosymbionts (Fig. 1A). We circumvented this limitation by successfully preserving the tissue architecture of the naturally expanded tentacle with a clamping technique marked using solid arrows in Figs. 1B and 1C. To verify the effectiveness of this technique, architectural differences in the gastrodermal tissue of extended and contracted tentacles were assessed by staining LBs with osmium tetroxide, cryosectioning fixed specimens, and imaging sections by using differential interference contrast microscopy (Figs. 1A & 1D) as described by Chen et al. (2012). This technique allowed the gastroderm to consist of a single cell layer (Fig. 1D).

Figure 1 Using capillary clips to fix tentacle-tissue architecture.

(A) Distribution of symbionts (Sym) and lipid bodies (LBs) in tentacles without capillary-clip fixation (scale = 10 µm). (B) Tentacles processed with (right) and without (left) the capillary clip (right; scale = one cm). (C) Comparison of tentacles with the capillary clip (solid arrow) with normal tentacle of Euphyllia glabrescens coral (open arrow; scale = one cm). (D) Distribution of Sym and LBs in tentacles with capillary-clip fixation (scale = 10 µm).

Ultrastructure analysis through Transmission Electron Microscope

Clamped tentacles were amputated using micro scissors and immediately fixed with 2.5% glutaraldehyde and 2% paraformaldehyde in 100 mM sodium phosphate containing 5% sucrose (pH 7.3) for 2.5 h at 4 ∘C then washed with 100 mM sodium phosphate at 4 ∘C. Specimens were post fixed in 1% osmium tetroxide in 50 mM sodium phosphate (pH 7.3) for 1 h at 4 ∘C. Samples and data were then proceeded and analyzed according to previously methods by Chen et al. (2012). Individual LB sizes were determined by measuring their maximum diameter (μm), and LBs were categorized into three size classes (micro-LBs: <1 μm; mid-LBs: 1–3 μm; large-LBs: >3 μm) and two distribution categories (near the mesoglea or near the coelenteron) by using Metamorph 6.3 (Molecular Devices).

Transcriptomic analysis of ER genes related to LB lipogenesis

Tentacle collection and RNA extraction

Tentacles (stretched to a length of ∼3 cm) were amputated from polyps of E. glabrescens colonies at approximately sunrise (06:00), noon (12:00), sunset (18:00), and midnight (00:00) using curved surgical scissors. The tentacles were then washed twice with filtered seawater (FSW), frozen in liquid nitrogen, and stored at −80 ∘C. De novo transcriptome sequencing (RNA-Seq) using Illumina technology was performed by Welgene Biotech (Taipei, Taiwan). Purified RNA was quantified at 260 nm (OD600) by using an ND-1000 spectrophotometer (Nanodrop Technology, Wilmington, DE, USA) and analyzed using a Bioanalyzer 2100 (Agilent Technologies, USA) with an RNA 6000 LabChip kit (Agilent Technologies, USA). All procedures were performed in accordance with manufacturer protocols.

Sequencing and transcriptome assembly

Libraries of all samples were created using the SureSelect Strand-Specific RNA Library Prep kit for 150 PE bp sequencing on the Solexa platform. Sequencing was performed using the TruSeq SBS Kit. Raw sequences were obtained using the Illumina Pipeline software bcl2fastq v2.0, and we expected to generate 2 Gb per sample. Trimmomatics was then used to generate qualified reads by trimming or removing low-quality bases or reads (QV ≥ 20) (Bolger, Lohse & Usadel, 2014). All qualified 150 PE reads were assembled using the de novo assembly program Trinity (Grabherr et al., 2011). After Trinity assembling, CD-HIT-EST was used to remove contig sequence redundancy (Fu et al., 2012). Unigene names were assigned by the partitioning algorithm in the khmer software package (Crusoe et al., 2015). Estimations of unigene abundance were calculated using Cufflinks.

Functional annotation of assembled unigenes

Unigenes in the transcriptome assembly were searched against the National Center for Biotechnology Information nonredundant protein (nr), Gene Ontology (GO), and Kyoto Encyclopedia of Genes and Genomes (KEGG) Pathway databases by using the high-efficiency alignment algorithm RAPSearch2 with a cutoff E value ≤ −3 (Zhao, Tang & Ye, 2011). The top alignment hits were used to predict the sequence orientations, GO accessions, and related KEGG Pathways of the unigenes. KEGG and GO functional classification for genes related to lipogenesis and LB formation was used to understand the distribution of gene functions in the coral host at the macro level. We used the fold change to assess the distributions and variations of genes related to LB lipogenesis for four sampling times.

Expression analysis of ER genes related to lipogenesis

Qualified 150 PE reads were mapped using the short-read alignment software Bowtie2 (Langmead & Salzberg, 2012). After mapping, the expression estimation of each gene was quantified using Cufflinks (Trapnell et al., 2012). Gene expression levels were calculated as fragments per kilobase of transcript per million mapped reads (FPKM). Furthermore, gene expression levels at different diel-cycle times were calculated as relative expression values of each sampling time. We adopted log2 (FPKM ratio) as the fold change of gene expression, that is, sunrise = log2(FPKM sunrise/midnight), noon = log2(FPKM noon/sunrise), sunset = log2(FPKM sunset/noon) and midnight = log2(FPKM midnight/sunset). FPKM and bioinformatic calculations were also performed by Welgene Biotech (Taipei, Taiwan). JMP v.10 (SAS Institute Inc., Cary, NC, USA) was used to translate the fold changes (log2 FPKM) of gene expression and the density distribution of gene expression profiles.

Diel expression pattern of BiP in host cells and purified LBs

Tentacle collection and LB isolation

Tentacles were sampled at six time points during the diel cycle: 06:00, 12:00, 15:00, 18:00, 21:00, and 00:00. For a given sampling time, a total of 80 tentacles were collected (pooled from multiple colonies). This sampling strategy was repeated for three distinct diel cycles (n = 3 different sampling days). After rinsing tentacles with FSW, tentacle tips were removed to prevent interference from nematocysts. The gastroderm was then separated from the epiderm through incubation with 3% N-acetylcysteine (pH 8.2, prepared in artificial seawater) for 1 h at room temperature (RT) (Peng et al., 2008). LBs samples and data were then proceeded and analyzed as previously described (Chen et al., 2012; Chen et al., 2017; Peng et al., 2011).

Fluorescence staining of purified LB

Lipid-specific fluorescent dye BODIPY 493/503 (Thermo Fisher Scientific) was prepared in ethanol as a 3.82 mM (1 mg/mL) stock. Each specimen was stained for 20 min with a working solution that was a 1:100 dilution of the stock (Gocze & Freeman, 1994). A fluorescein isothiocyanate (FITC) filter was used for green fluorescence imaging. The ER-specific staining dye ER-Tracker Blue-White DPX (Thermo Fisher Scientific) was prepared in a 1 mM dimethyl sulfoxide stock. LB specimens were stained for 1 h using a working solution that was a 1:250 dilution of the stock (Diwu et al., 1997). A 4′,6-diamidino-2-phenylindole filter and FITC filter were used for blue and green fluorescence, respectively.

Western blotting of BiP

Host gastrodermal cells were lysed, and LBs were delipidated in accordance with the procedures described by Mastro & Hall (1999) by adding a delipidation solution (tributyl phosphate: acetone: methanol, 1:12:1 v/v/v) to the collected fractions at a 14:1 v/v ratio on ice. This was followed by incubation at −20 ∘C overnight. Preprocessing of protein samples for SDS-PAGE analysis were proceeded as previously described in Peng et al. (2011).

Twenty micrograms of each protein sample were subjected to 12% SDS-PAGE then blotted onto polyvinylidene fluoride membranes (Immobilon-PSQ 0.45 mm; Millipore, Germany). The membranes were incubated in 5% skim milk in tris-buffered saline, 0.1% Tween-20 (TBST), 100 mM Tris (pH 7.6), and 150 mM NaCl at RT for 1 h. Incubation was then performed with rabbit anti-GRP78/BiP (ET-21) antibody (1:2000 dilution; cat. no G9043 Sigma-Aldrich) and preimmune serum antibodies (1:5000 dilution) in TBST buffer at 4 ∘C overnight. The membranes were then washed five times with TBST buffer for 10 min each, incubated with horseradish-peroxidase-conjugated goat anti-rabbit IgG antibodies (Millipore) in TBST buffer (1:5000 dilution), washed with TBST buffer, and visualized using a SuperSignal West Pico chemiluminescent substrate kit (cat. no 34080, Thermo Fisher Scientific) in accordance with manufacturer recommendations. Experiments were repeated in triplicate, and ImageJ was used to quantify band intensity for protein expression levels (Schneider, Rasband & Eliceiri, 2012). Relative BiP expression in the LB (or host) fraction was calculated as follows: the ratio of BiP in LB (or host) = the intensity quantification value in the LB (or host) fraction/the total intensity quantification value in the LB and host.

Statistical analysis

Data were analyzed using SPSS (version 14.0; SPSS Inc., Armonk, NY, USA). One-way analyses of variance and Tukey pairwise comparisons were conducted to determine the effect of sampling time on LB density, size, location, and gene and protein expression, and results were considered statistically significant when p was <0.05. Values are presented as means ± SDs.

Results

Ultrastructural and morphological characteristics of host LBs in the gastrodermis

Similar to the biogenesis of lipid droplets in other eukaryotic cells, that of LBs in coral–Symbiodiniaceae endosymbiosis has several unique morphological characteristics. We find multiple lines of evidence suggesting that lipid droplets are derived from the ER. First, coral host LBs were in close spatial proximity to mitochondria (Fig. 2), endoplasmic reticulum (ER, Fig. 2A), and Golgi apparatus (Fig. 2D and 2G), which exhibit high electron density upon transmission electron microscopy (TEM) examination. Our in situ morphology of coral host LBs revealed that they originate in the ER (Fig. 2A) and are surrounded by the ER (Fig. 2B). Figures 2B and 2C illustrates that micro-LBs and some mid-LBs accumulated in the coral host ER and then migrated and fused together. LBs formed a single globule with few inclusion bodies (Figs. 2C–2F) and had dense Golgi spots (Figs. 2D and 2G). When fusion occurred, numerous granules with high-electron density (Figs. 2E, 2F, 2H and 2I) appeared within the mid-LBs as they fused with micro-LBs, (Figs. 2E and 2H), and when mid-LB fused with large-LB (Figs. 2F and 2I). Finally, large-LBs continued to integrate into larger LBs.

Figure 2 Diel biosynthesis of host LBs in the gastroderm.

(A) LBs from the endoplasmic reticulum (ER, see colored areas, and different colours represent some fragments of ER) surrounded by mitochondria (red arrows; scale = 0.5 µm). (B) LBs enveloped by ER (solid red arrows; scale = 0.5 µm). (C) Fused LBs (scale = 0.5 µm). (D) LBs forming a single globule with several inclusion bodies (scale = 0.5 µm). (E) Single-globule LB continuing to fuse other micro-LBs with inclusion bodies (see solid yellow arrows, scale = 0.5 µm). High-electron density granules appeared within LBs. (F) Two mid-LBs fused together, and high-electron-density granules (see purple dashed frame) appeared within large-LBs (scale = 1 µm). (G) Enlargement of the red dashed frame in (D) displaying LB fusing near a Golgi apparatus and vesicles (see green dashed frame, scale = 0.2 µm). (H) Enlargement of the red dashed frame in (E). High-electron-density granules are visible between two fusing LBs (scale = 0.2 µm). (I) Enlargement of the red dashed frame in (F) (scale = 0.2 µm).

Diel pattern of LB size and distribution

To assess the spatiotemporal dynamics of coral LBs during the diel cycle, gastrodermal LBs were analyzed using TEM for three sizes (i.e., micro-LBs: <1 μm; mid-LBs: 1–3 μm; large-LBs: >3 μm) and two distributions near the mesoglea or coelenteron within the gastroderm (Fig. 3A). Total LBs started to increase at sunrise and gradually increased during the day, reaching maximum density at sunset (Fig. 3B).

Figure 3 LB size and distribution during the diel cycle.

(A) LB distribution was examined by calculating the number of LBs (red arrow) in two gastrodermal regions: the region near the mesoglea (mes) and the region near the coelenteron (coe; scale = 2 µm). (B) Photosynthetically active radiation and total LBs during a typical diel cycle. (C) Changes in size and distribution of LBs during the diel cycle. Data are presented as mean ±standard deviation (n = 3), * p < 0.05, ** p < 0.01, *** p < 0.001 (mesoglea vs. coelenteron; Table S1).

Micro-LBs comprised the largest portion of LBs throughout the diel cycle (Fig. 3C; Table S1). A significantly larger portion of micro-LBs were located near the mesoglea (approximately 40–80%) and they increased moderately throughout the diel cycle (Fig. 3C; Table S1; light period: 37.2 ± 2.9% to 44.3 ± 3.8%; dark period: 40.1 ± 3.4% to 53.3 ± 2.8%). The percentage of micro-LBs near the coelenteron gradually decreased from 22.3 ± 1.7% at sunrise to 10.0 ± 3.1% at sunset (Fig. 3C; Table S1) and increased overnight to 13.8 ± 1.6% at 21:00 and 20.3 ± 4.2% at 05:00. The number of mid- and large-LBs near the mesoglea remained relatively low (0.7 ± 0.3% and 13.5 ± 4.4%, respectively; Fig. 3C; Table S1) and nonsignificantly different during the diel cycle (mid-: p = 0.45; large-: p = 0.11). However, LB populations close to the coelenteron exhibited significant diel fluctuations (mid- and large-: p  <  0.05). The percentage of these larger LBs increased during the light period and returned to baseline levels overnight (Fig. 3C). The peak in relative abundance for mid- and large-LBs near the coelenteron occurred at 18:30 (23.3 ± 1.1%) and 21:00 (19.3 ± 1.7%), respectively.

Transcriptomic analysis of UPR activation/ER stress genes related to lipogenesis

In response to ER stress, a signal transduction pathway known as UPR is activated, and it may affect the metabolic process of de novo lipid biosynthesis. We analyzed the transcriptome of the coral Euphyllia glabrescens and focused on unigenes and pathways related to LB metabolism. LB lipogenesis occurs primarily in host cells, where metabolic signals regulate the expression of key enzymes in lipogenic pathways (Fig. 4A, modified from previous models; (Morris et al., 1997; Zheng, Zhang & Zhang, 2010; Basseri & Austin, 2012)). Specifically, as a monitor of UPR activation/ER stress, the ER chaperone and signaling regulator GRP78/BiP exhibited a fourfold increase in expression at noon. Nearly simultaneously, the two sensor proteins protein kinase R-like ER kinase (PERK) and inositol-requiring enzyme 1 (IRE1), which induce various pathways and drive lipid biosynthesis, exhibited near a two-fold increase in expression. IRE1 expression started to increase at sunrise and could reach a two-fold change in transcription factor X-box binding protein 1 (XBP1) expression by noon. The eukaryotic translation initiation factor 2 subunit 1 (eIF2 α) expression started to increase twofold change at sunrise. Transcription factor sterol regulatory element-binding protein 1 (SREBP1) activated by PERK also had increased expression at noon. The activating transcription factor 6 (ATF6) sensor protein was activated after midnight, whereas membrane-bound transcription factor site-1 protease (S1P) exhibited increased expression at sunset. Figure 4B presents a box plot of the distributions and outliers of gene expression for UPR activation/ER stress genes related to LB lipogenesis. Genes including BiP, PERK, eIF2α, SREBP1, IRE1, XBP1, ATF6, and S1P were upregulated starting at sunrise and downregulated from sunset to midnight.

Figure 4 Transcriptomic analysis of ER stress genes related to LB lipogenesis.

(A) ER stress activates the ER stress transducers PERK (left), IRE1 (center), and ATF6 (right) in the ER membrane. The orientation of the colored box corresponds to time points (i.e., sunrise, noon, sunset, and midnight) during the diel cycle. The color scale represents gene fold changes (log2 fragments per kilobase of transcript per million mapped reads) of gene expression (i.e., a red box represents a fourfold increase, and a green box represents a fourfold decrease). (B) Box plot of the distribution and outliers of gene expression levels for UPR activation/ER stress genes related to lipogenesis at sunrise, noon, sunset, and midnight.

Diel fluctuations of ER marker BiP locations

Figure 5A illustrates the successful fluorescence staining of purified LBs with lipid marker BODIPY 493/503 and ER-tracker Blue-White dye. The LBs contained neutral lipids and were enclosed in an ER membrane. The aforementioned in situ morphological findings also supported this finding (Fig. 2). BiP was selected to be an ER-specific marker protein to assess the functional association of coral gastrodermal LBs with the host ER. BiP expression was detected only in the coral host and purified LBs but not in endosymbionts (Fig. 5B). Furthermore, the 78-kDa BiP doublet was present in the purified LBs, indicating the close association of BiP with LBs. The two bands of BiPs in the host and purified LBs (Fig. 5B) were removed and analyzed using mass spectrometry (Fig. S1). The upper band in both LBs and the host was identified as a 78-kDa glucose-regulated protein (GRP78/BiP), and the lower band in the host fraction was identified as a 73-kDa heat shock protein (HSP70).

Figure 5 BiP expression during the diel cycle.

(A) [i] Lipid Bodies (LBs) purified from gastrodermal cells under differential interference contrast microscopy. [ii] LBs stained with the neutral lipid marker Bodipy 493/503. [iii] LBs stained with endoplasmic reticulum (ER)-tracker Blue-White dye, which indicated that LBs were enclosed in an ER membrane (scale = 10 µm). See Fig. S4 for different merging images. (B) BiP expression in the coral host (Host), dinoflagellate endosymbionts (Sym), and isolated LBs after Western blotting with the rabbit anti-BiP antibody. (C) BiP expression in the coral host (Host) and isolated LBs (LBs) during the diel cycle after Western blotting. (D) Further quantification and plotting of blots. See Figs. S2, S3 and Table S3 for the full-length blots and percentages of ER-related BiP expression in the host and LB fractions.

BiP expression during the diel cycle was assessed through Western blotting (Fig. 5C) and then quantified (Fig. 5D). To understand the relative ratios of BiP involved in LB lipogenesis of the total host fraction, the diel fluctuations in the relative concentration of BiP associated with purified LBs were assessed and compared with those of the remaining host fraction (Fig. 5D). The relative expression of BiP associated with LBs increased significantly at 15:00 and decreased at 18:00 (Fig. 5D). In addition, BiP abundance in the host fraction was significantly higher at night and peaked at 21:00. Thus, BiP exhibited distinct temporal fluctuation patterns in the LB and host fractions.

Discussion

Cytological characteristics of LB formation reveal LB origin

We successfully preserved the gastroderm consisting of a single layer of cells (Fig. 1). LBs move to self-assemble and fuse with each other (Boström et al., 2005). This is consistent with our observations that smaller LBs fuse to form larger LBs, resulting in irregularly shaped large LBs (Figs. 2B and 2C). The fusion of several smaller lipid droplets to larger lipid droplets also contributes to lipid droplet growth (Guo et al., 2008; Cheng, Fujita & Ohsaki, 2009). The LBs were in close spatial proximity to mitochondria, ER, and Golgi apparatus. Originating from the ER, lipid droplets can associate with most other cellular organelles through membrane contact sites (Jacquier et al., 2011; Peng et al., 2011; Olzmann & Carvalho, 2019). LB mobility was also attributed to the dynamic interactions of LBs with other organelles, including mitochondria (Figs. 2A, 2D, 2E and 2F), ER (Fig. 2A, Fig. 2B and Fig. 2C), the Golgi apparatus (Fig. 2D and 2G), peroxisomes, endosomes, and lysosomes (Herms et al., 2015; Gao & Goodman, 2015; Crossland, Barnes & Borowitzka, 1980; Hayes & Goreau, 1977; Vandermeulen, 1974). Therefore, LBs may play a role in the intracellular trafficking of lipids as well as proteins and other molecules between organelles. Moreover, LBs may migrate within host gastrodermal cells toward intracellular sites of lipid utilization and catabolism. Importantly, LBs biogenesis and degradation, as well as their interactions with other organelles, are tightly coupled to cellular metabolism and facilitate the coordination and communication between different organelles and act as vital hubs of cellular metabolism (Olzmann & Carvalho, 2019). In many cells, lipid droplets undergo active motion, typically along microtubules. This motion has been proposed to aid growth and breakdown of droplets, to allow net transfer of nutrients from sites of synthesis to sites of need and to deliver proteins and lipophilic signals (Pol et al., 2004; Welte, 2009). Thus, nascent LB globule biogenesis likely occurs near the mesoglea. As LBs grow and mature, they migrate across gastrodermal cells and fuse to form large LBs adjacent to the coelenteron.

The time-dependent LB relocation may occur inside a single gastrodermal cell during their biogenesis which ultimately manifests as a redistribution of LBs across the gastrodermal tissue layer. LBs move along microtubules and relocate within the cytoplasm (Welte, 2009). A proteomic investigation indicated that cytoskeletal proteins and proteins involved in intracellular trafficking were associated with coral LBs (Peng et al., 2011). The mechanism by which LB populations migrate across the gastroderm to cause this redistribution remains unknown. By correlating LB size and location throughout the diel cycle in naturally expanded tentacle specimens, a study validated the proposed redistribution of LBs from the mesoglea to the coelenteron (Chen et al., 2012). Our data further demonstrated that micro- and mid-sized LBs fuse to form large LBs near the coelenteron. In particular, micro-LBs were the most abundant near the mesoglea, whereas mid- and large-sized LBs increased in abundance near the coelenteron during the light period (Fig. 3). This finding was consistent with a previous study suggesting that LBs migrate across the gastroderm during the diel cycle (Chen et al., 2012).

Transcriptional variation of BiP is correlated with LB lipogenesis

Ultrastructural morphology studies have revealed that lipid droplets are closely associated with the ER membrane (Fig. 2), and proteomic studies of lipid droplets isolated from various cell lines have revealed numerous ER proteins (Chen et al., 2012; Peng et al., 2011; Liu et al., 2004; Brasaemle et al., 2004; Wan et al., 2007). The ER is the main site of the synthesis of lipids that constitute most of the lipid components of all biological membranes, and the ER is highly sensitive to changes in intracellular homeostasis and extracellular stimuli. During the perturbation of ER homeostasis, referred to as ER stress, UPR is activated when misfolded proteins accumulate, when reactive oxygen species (ROS) are produced, or when the accumulation of FAs resulting from lipolysis alter ER membrane lipids (Volmer & Ron, 2015; Walter & Ron, 2011; Chitraju et al., 2017). We demonstrated that the increase in BiP gene expression from sunrise to noon (Fig. 4) may reflect a localized increase in BiP in daytime within the ER microdomain involved in LB formation (Fig. 5). BiP is the primary regulator of the UPR and is upregulated in response to ER stress (Bertolotti et al., 2000; Morris et al., 1997). Furthermore, PERK and IRE1 cascade facilitates LB formation and possibly lipogenesis, and the ATF6 branch enhances phospholipid biosynthesis (Zheng, Zhang & Zhang, 2010; Basseri & Austin, 2012; Zha & Zhou, 2012). Our transcriptiomic variation results (Fig. 4B) were also consistent with those of previous studies, which indicated a general increase in total lipids, FA concentration, and density in LBs during the light period followed by a decrease at night (Chen et al., 2012; Chen et al., 2017).

In addition to its core function in lipid formation, the diel cycle may provide a mechanism for managing daily oxidative stress induced by photosynthesizing Symbiodinaceae in coral host tissue (Levy et al., 2006). In addition to ROS production in the ER resulting from oxidative protein folding, this could overwhelm ER homeostasis and induce daily ER stress response (Malhotra & Kaufman, 2007; Levy et al., 2011; Oakley et al., 2017). LB formation may play a role in alleviating ER stress by sequestering unfolded proteins and transiently storing them for degradation (Ploegh, 2007). This has been supported by the observations of increased LB biogenesis induced by ER stress (Fei et al., 2009; Lee et al., 2012), but this functional link remains uncertain (To et al., 2017).

Translational fluctuations of BiP revealed LB origins

After analyzing diel transcriptional variations and the transcriptomic pathway, we selected the ER marker BiP associated with LBs to elucidate the connection among LB formation, the ER, and the diel cycle. We observed an increase in the relative concentration of BiP associated with LBs in midafternoon (Fig. 5). However, BiP expression in the host fraction continued to increase at night (21:00). The increase in the apparent relative abundance of BiP may be due to an increase in either ER network volume or localized BiP concentration in response to upregulated BiP expression. Our current data cannot be used to distinguish one cause from the other. Nevertheless, the consistent association of BiP with LBs throughout the diel cycle and the diel patterns in BiP concentrations between the LB and host fractions demonstrated that LBs are closely associated with a specialized domain of the host ER. The peak in LB-associated BiP was associated with an increase in the total volume of LBs (Fig. 5), thereby supporting the involvement of ER expansion in LB biosynthesis. Moreover, the nocturnal increase in host ER BiP but not in LB formation may indicate ER expansion at night. The ER is a dynamic organelle, and its tubular network can expand to meet the demands of ER-based processes, including protein folding, modification, and secretion; lipid metabolism; and calcium homeostasis (Federovitch, Ron & Hampton, 2005).

A persisting question in LB research is whether LBs ever truly detach from the ER (Mishra et al., 2016). This is particularly relevant considering the substantial migration and fusion of LBs in the coral gastroderm. After the peak in LB-associated BiP occurred in the midafternoon, the relative BiP concentration decreased at sunset (Fig. 5D). However, total LB volume reaches its highest concentration at sunset (Chen et al., 2012; Chen et al., 2017; Peng et al., 2011). Therefore, larger LBs that dominate at sunset likely detach from the ER or are at least less closely associated with the ER. The continuity of the ER membrane with the LB surface may be maintained permanently with stalk-like projections of the ER membrane that allow for the continued lipid and protein trafficking between the organelles (Jacquier et al., 2011; Zehmer et al., 2009). Moreover, the dynamic fluctuation of BiP expression between the coral host and LBs rules out the possibility that BiP within the LBs was merely an impurity collected in the isolation process. Rather, our findings suggested that coral gastrodermal LBs are functionally connected to a specialized domain of the host ER.

Dynamic schematic model of LB formation

Coral gastrodermal LBs are highly dynamic organelles, and LB formation is highly associated with coral host ER. Imaging analyses of ultrastructural morphological characteristics and the Western blotting of BiP–ER marker protein distribution supported the model of LB formation (Fig. 6). We developed this model of ER involvement in LB biogenesis to illustrate the distribution of BiP expression between a coral host and LBs (Fig. 6A), resulting in dynamic fluctuations in BiP expression (Fig. 5). Furthermore, we proposed a six-stage process of LB formation and maturation (Fig. 6B). First, micro-LBs bud from the ER surrounded by mitochondria (Fig. 6B [i]). Subsequently, more micro-LBs gather and are surrounded by ER sheaths (Fig. 6B [ii]). In the third stage, micro-LBs fuse into larger LBs with different lipid compositions (Fig. 6B [iii]). This might result in different colors in TEM image analysis. During the diel cycle, LB growth and fusion result in mid-LBs. At the same time, a large population of micro-LBs is maintained (Fig. 6B [iv]); this occurs with dense Golgi spots in particular. Mid-LBs continue to fuse to form large-sized LBs (>6 μm). When fusion occurs, numerous “dots” form on vesicles in between the LBs, as illustrated in Fig. 6 [v]. Finally, the large-LBs fuse to form mature LBs, as presented in Fig. 6 [vi].

Figure 6 Schematic summary of LB formation in coral–Symbiodinaceae endosymbiosis.

(A) Proposed model of endoplasmic reticulum (ER) involvement in LB biogenesis resulting in the BiP expression distribution between coral host and LBs. (B) LB formation and maturation. [i] LB budding. [ii] ER surrounding. [iii] LB assembling. [iv] Golgi modification. [v] LB fusion. [vi] LB maturation.

LB size must be strictly regulated to balance lipid storage and mobilization in response to energetic demand (Yang et al., 2012). In response to excess lipid molecules, a cell can increase either the number of LBs or the size of existing LBs. Changes to LB size alter the LB volume-to-surface-area ratio, which affects the rate of LB catabolism. The formation of large LBs as observed in the coral gastroderm (Fig. 3C) is the most efficient form of lipid storage (Yang et al., 2012). Moreover, micro-LBs potentially function to meet immediate energetic demands because of the rapid availability of lipids for catabolism and oxidation (Ariotti et al., 2012). Therefore, heterogeneity in gastrodermal LBs may reflect diversity in LB functions (Zhang et al., 2016). The overnight disappearance of large LBs was hypothesized to occur because of nocturnal lipid catabolism and utilization (Chen et al., 2012), however, the underlying mechanism is not clear.

Conclusions

Unique diel rhythmicity in LB biosynthesis, growth, and maturation was observed in coral gastrodermal cells, and these processes may play a key role in regulating coral–Symbiodinaceae endosymbiosis (Chen et al., 2012; Chen et al., 2017; Peng et al., 2011). The daily cycle of LB formation, maturation, and degradation is a cellular process underpinning stable coral–Symbiodinaceae endosymbiosis (Chen et al., 2012; Peng et al., 2011). By demonstrating that LB biogenesis is localized in the host ER, this study provided new insight into the mechanism of metabolite integration within this symbiosis. LBs are composed of FA moieties with both host and symbiont origins (Chen et al., 2017). Therefore, lipid molecules translocated from the symbiont to the host may be transported to the host ER where they enter storage lipid biosynthetic pathways in the host and become integrated into LBs.

In conclusion, the ER plays a central role in endosymbiosis regulation. Our results demonstrated that LBs originate in specialized domains of the host ER (likely localized near the mesoglea) but disassociate from the ER when migrating and fusing to form large LBs near the coelenteron. Consequently, we predicted that the disruption of host ER homeostasis induced by environmental stress can damage endosymbiotic function (Oakley et al., 2017). Further research is required to elucidate the precise mechanisms of LB formation, maturation, and catabolism in corals and to examine the roles, apart from energy homeostasis, that LBs might have in coral–Symbiodinaceae endosymbiosis.

Supplemental Information

Supplemental Information 1 Raw images with full-length gels and blots and raw LC-MS/MS data of BiP protein

Click here for additional data file.

Supplemental Information 2 Raw data: Transcriptomic sequences of ER stress genes related to LB lipogenesis

Click here for additional data file.

Supplemental Information 3 Raw data - Fig. 3B & C

Click here for additional data file.

Supplemental Information 4 Raw data - Fig. 4B

Click here for additional data file.

Supplemental Information 5 Raw data - Fig. 5D

Click here for additional data file.

We thank Yi-Jyun Chen for performing the experiments. We thank Dr. Anderson B. Mayfield and Crystal J. McRae for reviewing the manuscript draft. We thank the anonymous reviewers whose comments/suggestions helped improve and clarify this manuscript.

Additional Information and Declarations

Competing Interests

Author Contributions

DNA Deposition

Data Availability

The authors declare there are no competing interests.

Hung-Kai Chen conceived and designed the experiments, performed the experiments, analyzed the data, prepared figures and/or tables, authored or reviewed drafts of the paper, and approved the final draft.

Sabrina L Rosset analyzed the data, prepared figures and/or tables, and approved the final draft.

Li-Hsueh Wang performed the experiments, analyzed the data, prepared figures and/or tables, and approved the final draft.

Chii-Shiarng Chen conceived and designed the experiments, authored or reviewed drafts of the paper, and approved the final draft.

The following information was supplied regarding the deposition of DNA sequences:

The sequences are available at GenBank: MW659935 to MW659942. The reference table and the sequences are available in the Supplementary File.

The following information was supplied regarding data availability:

The raw images with full-length gels and blots, raw LC-MS/MS data of BiP protein, and transcriptomic sequences of ER stress genes related to LB lipogenesis are available in the Supplemental Files.

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
