# Peer review of "The characteristics of host lipid body biogenesis during coral-dinoflagellate endosymbiosis"

_PeerJ, doi:10.7717/peerj.11652_

## Round 0.1 · original submission · Major Revisions

Dear Authors,
As you can see the manuscript has been evaluated by 3 reviewers.
I encourage you to resubmit the manuscript as all three find the work to be interesting.

Please see comments mainly from reviewers 2 & 3.

·

Basic reporting

This is a well planned experimental work putting together an interesting biological model-coral-dinoflagellate endosymbiosis-with the use of a series of morphological and molecular techniques. The manuscript is well written and illustrated, with results that contribute to a better knowledge of lipid biogenesis in a natural in vivo context.

Experimental design

The work was well planned and executed. The materials and methods sections details all basic information on coral husbandry, light and electron microscopy analysis of the lipid bodies, transcriptomic analysis related to genes involved in the lipid body biogenesis as well as from the endoplasmic reticulum and bioinformatic analysis.

Validity of the findings

The results obtained shows clearly the lipid body formation in the gastrodermal tissue layer of the coral, the rhythmicity in the size and distribution of lipid bodies. It is convincingly shown that the lipid bodies are formed in the endoplasmic reticulum, as previous shown in other cell types. Transcriptomic analysis support the morphological observations.

Additional comments

Congratulations for this interesting piece of work where an excellent natural in vivo model was used to follow the biosynthesis of lipid bodies.

Reviewer 2 ·

Basic reporting

The paper is well writen throught, the structure is clear in general, although there are a few weaknesses in the figures and captions (please see "comments").

Experimental design

Research well design. Methods explained in enough detail. Please see "comments".

Validity of the findings

Please see comments section.

Additional comments

The paper by Chen et al examines lipid body content, size and localisation in the host cells of symbiotic corals (in the species Euphyllia glabrescens). The authors have also studied the diel cycle dynamic in the distribution of lipid bodies across the coral host gastrodermal region, and differential regulation of genes and enzyme potential involved in lipogenesis in E. glabrescens.
This paper is topical and the authors have done careful experiments, including improving their protocol for collection of tentacle samples for TEM analysis. Overall, I think this work provides interesting elements to forward our understanding of lipid body formation, daily dynamic and potentially lipid synthesis.
I would suggest, however, that a few aspects that need to be better clarified before publication of this work:
- Lines 232-236 and Figure 1: Although I appreciate the improvement of the method that the authors have achieved to examine the ultrastructure of the tentacle by the clamping technique I do not think that Figure 1 is justified as part of the results section. This display and related description should be better placed in the Supplemental section.
- I find that the photographs shown in Figure 2 do not unequivocally support the statements made in lines 237-246, particularly because it is not really possible to discern many features from the displays (for example in panels a and b). The authors should in the first place clarify how they come to the conclusion that the structures that they point to are in fact lipid bodies. This is critical for the further revision of the rest of the manuscript, which is strongly based on a potential close associated between the LB and the RE. For instance, the discussion of potential interaction with other intracellular structures (lines 313-321) should be revised accordingly to clarify how that is extrapolated from the data shown in Figure 2. I wonder if clarity in this figure can be at least partially improved by (i) labelling other components in the photographs and (ii) positioning the arrows more specifically (particularly for example the identification of LBs in panels a and b are confusing).
- I struggled to see why the authors explain the expression of BiP protein as a response to ER- stress. As I see from the M&M the corals were kept under control environmental conditions and corals should be fully acclimated to the light environment. Is the argument that the daily fluctuations in ROS due to the photosynthesis of the symbiont results in ER stress within the host cells also in fully acclimated corals? Can the authors explain that more in detail?
- Figure 5: Could the authors clarify what are the different sizes of the LBs that the images show are? Is there any correspondence of these to the LB sizes that were classified in Figure 3 or are the different sizes a result of the purification step? I’m trying to understand the results presented since in panel Ai there seems to be more “LB” than are recognised by either the lipid marker (Aii) or the ER-tracker (Aiii). Also, panel D in that figure is not clear.
- Apologies if it is my omission but I could not find where the transcriptome data has been deposited.

Reviewer 3 ·

Basic reporting

The report matched categories for basic reporting.

Experimental design

The report matched categories for experimental design.

Validity of the findings

I only have some concerns on the general interpretation & presentation of the data, which are included in the "general comments for the authors".

Additional comments

The paper presents an interesting phenomenon, in which the formation of lipid body in coral during coral-dinoflagellate endosymbiosis, represented by LB size and distributions fluctuate by diel rhythmicity. The study also revealed critical transcriptomic regulation of lipid biosynthesis. The experiments are well-executed, with experimental procedures described in much detail in the Methods section. However, some concerns about the results need to be addressed before this manuscript can be accepted for publication in PeerJ.

Line 254: Please specify the method to count the LB on mesoglea side or coelenteron side (What percentage of the LB falling into a certain side is considered counted as mesoglea or coelenteron?)

Line 277-280: On Figure 4, it appears that PERK and IRE gene expression increased by one log, which means 2 folds, not one fold as stated in this sentence.

I encourage the author to deposit the data of their RNA-Seq analysis. The supplementary data only contained a small number of genes (BIP, PERK, IRE1, etc.). A GO analysis of the RNA-Seq data could reveal even more patterns in gene expression during diel fluctuations.

Figure 5A: the author should perform an analysis of colocalization between Bodipy and ER Tracker signals. This can be done with ImageJ. A merged image of these 2 channels should also be included.

The authors claimed micro-LBs fused to form larger LBs. I assumed that
However, looking at micro-LB, can the authors provide an explaination why the number of Micro-LB did not really change much between light to dark phase. (a one-way ANOVA analysis of samples near mesoglea would be able to distinguish if there is any difference in Micro-LB number over time)

Figure 2: Panels in the second and third columns appeared to be at different scale compated to the corresponding rows in the first column. For example: 2I scale bar is larger than 2G,H even though they are at the same magnification. Working with images at the inaccurate scale could lead to inaccurate quantification of the LB size. I encourage the authors to double check their raw data/images to confirm this.

Figure 5B: what was the time-point chosen for this Western Blot?

---

## Round 0.2 · accepted · Accept

The authors have answers nicely all concern raised by the reviewers.

---

## Author Rebuttal · Round 0.2

Planning & Research Division

**National Museum of Marine Biology and Aquarium**

2 Houwan Road Checheng Pingtung 94450 Taiwan

Tel: +886-8825001 ext. 1361

Fax: +886-8825087

https://www.nmmba.gov.tw

cchen@nmmba.gov.tw

May 20th, 2021

Dear Prof. Levy,

We appreciate the constructive comments from you and the anonymous reviewers on the previous version of our manuscript entitled "The characteristics of host lipid body biogenesis during coral-dinoflagellate endosymbiosis" (PeerJ-2021:02:58163:1:0). We have revised our manuscript (including newly presented supplementary data) to address the issues raised by the reviewers.

We have made several revisions to our manuscript in response to reviewers' comments. First, according to the suggestions of both reviewers, we have moved the text referring to Figure 1 from the Results section to the Materials and Methods section. Secondly, we have added clarifying labeling to key features on Figure 2 to clearly highlight areas of interest. Finally, we have also attached an additional supplementary figure to address the concerns of reviewers #2 and 3 in the "point-by-point" response to the reviewers' comments. These additional data will not appear in the main manuscript but should hopefully give reviewers more confidence in our findings.

We hope that these thorough revisions to our manuscript are to your satisfaction and that it is now suitable for publication in PeerJ. Please do not hesitate to contact me if any additional data and/or information could be provided that would expedite the rapid publishing of this work.

Thank you again for your consideration of our manuscript for publication in PeerJ.

Sincerely,

陳啟祥

Prof. Chii-Shiarng Chen, Ph.D. (corresponding author)

Director General, National Museum of Marine Biology and Aquarium, Taiwan, ROC

# *Point-by-Point response to reviewers' comments on manuscript Submission ID PeerJ-2021:02:58163:1:0*

## Reviewer #1 (Wanderley de souza)

### Basic reporting

*This is a well planned experimental work putting together an interesting biological model-coral-dinoflagellate endosymbiosis-with the use of a series of morphological and molecular techniques. The manuscript is well written and illustrated, with results that contribute to a better knowledge of lipid biogenesis in a natural in vivo context.*

### Experimental design

*The work was well planned and executed. The materials and methods sections details all basic information on coral husbandry, light and electron microscopy analysis of the lipid bodies, transcriptomic analysis related to genes involved in the lipid body biogenesis as well as from the endoplasmic reticulum and bioinformatic analysis.*

### Validity of the findings

*The results obtained shows clearly the lipid body formation in the gastrodermal tissue layer of the coral, the rhythmicity in the size and distribution of lipid bodies. It is convincingly shown that the lipid bodies are formed in the endoplasmic reticulum, as previous shown in other cell types. Transcriptomic analysis support the morphological observations.*

### Comments for the author

*Congratulations for this interesting piece of work where an excellent natural in vivo model was used to follow the biosynthesis of lipid bodies.*

### Author's response:

We appreciate the reviewer's positive comments.

## *Reviewer #2:*

### *Basic reporting*

*The paper is well written throught, the structure is clear in general, although there are a few weaknesses in the figures and captions (please see "comments").*

### *Experimental design*

*Research well design. Methods explained in enough detail. Please see "comments".*

### *Validity of the findings*

*Please see comments section.*

### *Comments for the author*

*The paper by Chen et al examines lipid body content, size and localisation in the host cells of symbiotic corals (in the species Euphyllia glabrescens). The authors have also studied the diel cycle dynamic in the distribution of lipid bodies across the coral host gastrodermal region, and differential regulation of genes and enzyme potential involved in lipogenesis in E. glabrescens. This paper is topical and the authors have done careful experiments, including improving their protocol for collection of tentacle samples for TEM analysis. Overall, I think this work provides interesting elements to forward our understanding of lipid body formation, daily dynamic and potentially lipid synthesis.*

*I would suggest, however, that a few aspects that need to be better clarified before publication of this work:*

1. *Lines 232-236 and Figure 1: Although I appreciate the improvement of the method that the authors have achieved to examine the ultrastructure of the tentacle by the clamping technique I do not think that Figure 1 is justified as part of the results section. This display and related description should be better placed in the Supplemental section.*

### Author's response:

We appreciate the reviewer's comments. Since we described the tentacle clamping technique in the **"Materials and Methods" section**, we have now moved the text referring to "**Figure 1**" from the **"Results"** section to **"Materials and Methods".** The text has now been revised in the manuscript accordingly, as per your suggestions. Specifically, we have changed the pertinent sentences in the **"Materials and Methods"** and **"Results"** sections: lines 103-113, and lines 237-241.

*2.  I find that the photographs shown in Figure 2 do not unequivocally support the statements made in lines 237-246, particularly because it is not really possible to discern many features from the displays (for example in panels a and b). The authors should in the first place clarify how they come to the conclusion that the structures that they point to are in fact lipid bodies. This is critical for the further revision of the rest of the manuscript, which is strongly based on a potential close associated between the LB and the RE. For instance, the discussion of potential interaction with other intracellular structures (lines 313-321) should be revised accordingly to clarify how that is extrapolated from the data shown in Figure 2. I wonder if clarity in this figure can be at least partially improved by (i) labelling other components in the photographs and (ii) positioning the arrows more specifically (particularly for example the identification of LBs in panels a and b are confusing).*

**Author's response:**

As shown in previous study, mitochondria, endoplasmic reticulum (ER), and Golgi apparatus were commonly observed around the LB [see "**Reference list for reviewer #2**" **Reference 1**]. In comparison with lipid droplets of other eukaryotic cells, there are several unique morphological characteristics of LBs in this coral–dinoflagellate endosymbiosis. Our in situ morphology of coral host LBs revealed that the major core components are different from those of lipid droplets in other eukaryotic cells. For example, numbers of unsaturated bonds in triacylglycerols resulting from feeding cells with fatty acids have been shown to determine the TEM electron density of lipid droplets [see "**Reference list for reviewer #2**" **Reference 2**]. Based on the high reactivity of osmium tetroxide with unsaturated fatty acids, an increase in the electron density of the lipid droplets as examined by quantitative TEM imaging analysis was observed when more unsaturated fatty acids were fed to both fibroblasts and adipocytes. The electron density of lipid droplets varies across cell types, ranging from low (e.g., adipocytes) to high (e.g., leukocytes). The LBs in *E. glabrescens* are more similar to the latter in that they are characterized by a high electron density suggestive of a high level of unsaturated triacylglycerols. This also confirmed that, among other lipids, triacylglycerols are most abundant in both weight concentration and molecular number in coral LBs [see "**Reference list for reviewer #2**" **Reference 1**].

We also found other distinguishing characteristics: sections of the LBs exhibited rectangular, electron-transparent inclusions, measuring 1–2 µm in length and 25–30 nm in width. The occurrence of these inclusions varied in the LBs examined. The nature of these inclusions remains to be elucidated. However, they may represent regions containing lipid species without unsaturated bonds or may not be reactive with osmium tetroxide such as sterols and wax esters. As previously shown, the electron density of cholesterol-rich lipid droplets was generally low [see "**Reference list for reviewer #2**" **Reference 2**].

We have added labels to **"Figure 2"** to enhance clarity. Previously unclear descriptions of key features in **"Figure 2"** are now labeled, this includes: **mitochondria in panel a, d, e and f (see red arrows)**, **endoplasmic reticulum (ER) in panel a, b and c (see colored areas, and different colours represent some fragments of ER)**, **Golgi apparatus in panel d and g (see green dashed frame)**, **inclusion bodies in panel c, d, e and f (see yellow arrows)** [see "**Reference list for reviewer #2**" **Reference 1, 2**], **high-electron density granules in panel e, f, h and i (see purple dashed frame)** [see "**Reference list for reviewer #2**" **Reference 1, 2**]. We have revised the text, figure caption, and legend accordingly:

*Lines 242-254 of "Results" section:* "Similar to the biogenesis of lipid droplets in other eukaryotic cells, that of LBs in coral–Symbiodiniaceae endosymbiosis has several unique morphological characteristics. We find multiple lines of evidence suggesting that lipid droplets are derived from the ER. First, coral host LBs were in close spatial proximity to mitochondria (Figure 2), endoplasmic reticulum (ER, Figure 2A), and Golgi apparatus (Figure 2D and G), which exhibit high electron density upon transmission electron microscopy (TEM) examination. Our in situ morphology of coral host LBs revealed that they originate in the ER (Fig. 2A) and are surrounded by the ER (Fig. 2B). Fig. 2B and C illustrate that micro-LBs and some mid-LBs accumulated in the coral host ER and then migrated and fused together. LBs formed a single globule with few inclusion bodies (Figure 2C, D, E and F) and had dense Golgi spots (Fig. 2D and 2G). When fusion occurred, numerous granules with high-electron density (Figure 2E, F, H and I) appeared within the mid-LBs as they fused with micro-LBs, (Fig. 2E and 2H), and when mid-LB fused with large-LB (Fig. 2F and 2I). Finally, large-LBs continued to integrate into larger LBs."

*Lines 321-341 of "Discussion" section:* "This is consistent with our observations that smaller LBs fuse to form larger LBs, resulting in irregularly shaped

large LBs (Fig. 2B and C). The fusion of several smaller lipid droplets to larger lipid droplets also contributes to lipid droplet growth (Guo et al., 2008; Cheng *et al.*, 2009). The LBs were in close spatial proximity to mitochondria, ER, and Golgi apparatus. Originating from the ER, lipid droplets can associate with most other cellular organelles through membrane contact sites (Jacquier *et al.*, 2011; Peng *et al.*, 2011; Olzmann & Carvalho, 2019). LB mobility was also attributed to the dynamic interactions of LBs with other organelles, including mitochondria (Fig. 2A, D, E and F), ER (Fig. 2A, B and C), the Golgi apparatus (Fig. 2D and 2G), peroxisomes, endosomes, and lysosomes (Herms *et al.*, 2015; Gao & Goodman, 2015; Crossland, Barnes & Borowitzka, 1980; Hayes & Goreau, 1977; Vandermeulen, 1974). Therefore, LBs may play a role in the intracellular trafficking of lipids as well as proteins and other molecules between organelles. Moreover, LBs may migrate within host gastrodermal cells toward intracellular sites of lipid utilization and catabolism. Importantly, LBs biogenesis and degradation, as well as their interactions with other organelles, are tightly coupled to cellular metabolism and facilitate the coordination and communication between different organelles and act as vital hubs of cellular metabolism (Olzmann & Carvalho, 2019). In many cells, lipid droplets undergo active motion, typically along microtubules. This motion has been proposed to aid growth and breakdown of droplets, to allow net transfer of nutrients from sites of synthesis to sites of need and to deliver proteins and lipophilic signals (Pol *et al.*, 2004; Welte, 2009). Thus, nascent LB globule biogenesis likely occurs near the mesoglea. As LBs grow and mature, they migrate across gastrodermal cells and fuse to form large LBs adjacent to the coelenteron."

3. *I struggled to see why the authors explain the expression of BiP protein as a response to ER- stress. As I see from the M&M the corals were kept under control environmental conditions and corals should be fully acclimated to the light environment. Is the argument that the daily fluctuations in ROS due to the photosynthesis of the symbiont results in ER stress within the host cells also in fully acclimated corals? Can the authors explain that more in detail?*

**Author's response:**

BiP, also referred to as GRP78, is a major endoplasmic reticulum (ER) chaperone protein critical for protein quality control of the ER, a master regulator for

ER stress due to its role as a major ER chaperone with anti-apoptotic properties as well as its ability to control the activation of transmembrane ER stress sensors (IRE1, PERK, and ATF6) through a binding-release mechanism. [see "**Reference list for reviewer #2**" **References 4-5**].

The abundance of Symbiodiniaceae is important because it may directly affect the amount of oxygen produced within corals cells. While endosymbiont photosynthesis serves as the engine to power the growth of coral reefs, sunlight capture, absorption, and utilization presents a high potential for photo-oxidative damage. Oxidative stress results from the production and accumulation of reactive oxygen species (ROS). The photosynthesis invariably produces for ROS, are common by-products of normal aerobic cell metabolism and high level of ROS can damage lipids, proteins, and DNA, as well as induce the emergence of unfolded or misfolded proteins to both coral host and symbiont tissues. At the cellular scale, the radiative exposure of Symbiodiniaceae cells, resulting in photodamage and the subsequent generation of ROS that influence the coral symbiosis. Furthermore, the ROS of Symbiodiniaceae also will affect the metabolism of coral host cell during the coral-Symbiodiniaceae endosymbiosis. These unfolded or misfolded proteins would accumulate in the lumen of the ER resulting imbalances of ER homeostasis and activate the unfolded protein response. [see "**Reference list for reviewer #2**" **References 6-10**].

Our coral culture facility has a similar light intensity and photoperiod as nearby natural reefs [see "**Reference list for reviewer #2**" **Reference 11** for details on our aquarium light levels]. Typical natural reef conditions in southern Taiwan are well documented [see **"Reference list for reviewer #2" Reference 12**], particularly at the sites from which corals were collected, and seawater temperature fluctuates from 20-30°C annually [see **"Reference list for reviewer #2" Reference 13 and 14**]. The mean annual temperature in Nanwan Bay, the site from which corals were collected, is 26°C. We aimed to maintain our corals at a temperature similar to in situ conditions throughout our experiments. Coral colonies were maintained at 26.5 ± 2°C using a cooler under a natural photoperiod.

4.   *Figure 5: Could the authors clarify what are the different sizes of the LBs that the images show are? Is there any correspondence of these to the LB sizes that were classified in Figure 3 or are the different sizes a result of the purification*

*step? I'm trying to understand the results presented since in panel Ai there seems to be more "LB" than are recognised by either the lipid marker (Aii) or the ER-tracker (Aiii). Also, panel D in that figure is not clear.*

**Author's response:**

The different sizes were the result of the purification step. As described in the "**Materials and Methods**" section, we separated the gastroderm tissue of coral tentacles from the epiderm through incubation with 3% *N*-acetylcysteine. After tissue layer separation, a procedure was developed to purify LBs directly from the symbiotic gastrodermal cells (SGCs) by two cycles of subcellular fractionation and detergent washes. This process starts with LB isolation from the gastroderm; first, homogenization is achieved through a glass tissue grinder followed by a syringe to break host gastrodermal cells (but not Symbiodiniaceae) and release only host LBs. Different sizes of LBs fractions were isolated from the remaining host fraction by sucrose gradients as described in previous studies [see "**Reference list for reviewer #2**" **References 1, 15 and 16**].

We appreciate the reviewer's valuable comments and we have included new merged images of Bodipy and ER Tracker channels for reference; please see "**Appendix figures for reviewer #2**" **Appendix Figure 1**]. We have also enhanced the images in Figure 5 and added these to "**Supplementary Figure S4.**

The panel D of Figure 5 shows the relative BiP expression in the LB (or host) fraction which was calculated as follows: the ratio of BiP in LB (or host) = the intensity quantification value in the LB (or host) fraction/the total intensity quantification value in the LB and host.

5.  *Apologies if it is my omission but I could not find where the transcriptome data has been deposited.*

**Author's response:**

The sequences described here are accessible via GenBank, access numbers MW659935 to MW659942. The reference table and sequences were upload as Supplementary Files 2. The link for the third party database is

https://www.ncbi.nlm.nih.gov/WebSub/?form=dwnld&sid=2432566&tool=genbank.

Note: you need to log in with an account on the NCBI website, and then click the link above to download the file.

## *Reference list for reviewer #2*

1. Peng SE, Chen WNU, Chen HK, Lu CY, Mayfield AB, Fang LS, and Chen CS. 2011. Lipid bodies in coral-dinoflagellate endosymbiosis: proteomic and ultrastructural studies. Proteomics 11(17): 3540-3555 DOI 10.1002/pmic.201000552.

2. Cheng J, Fujita A, and Ohsaki Y. 2009. Quantitative electron microscopy shows uniform incorporation of triglycerides into existing lipid droplets. Histochemistry and Cell Biology 132: 281-291 DOI 10.1007/s00418-009-0615-z.

3. Olzmann JA, and Carvalho P. 2019. Dynamics and functions of lipid droplets. Nature reviews Molecular cell biology 20(3): 137-155 DOI 10.1038/s41580-018-0085-z.

4. Lee AS. 2005. The ER chaperone and signaling regulator GRP78/BiP as a monitor of endoplasmic reticulum stress. Methods 35(4): 373-381 DOI 10.1016/j.ymeth.2004.10.010.

5. Wang M, Wey S, Zhang Y, Ye R, and Lee AS. 2009. Role of the unfolded protein response regulator GRP78/BiP in development, cancer, and neurological disorders. Antioxidants & redox signaling 11(9): 2307-2316 DOI 10.1089/ars.2009.2485.

6. Franklin DJ, Hoegh-Guldberg O, Jones RJ, and Berges JA. 2004. Cell death and degeneration in the symbiotic dinoflagellates of the coral *Stylophora pistillata* during bleaching. Marine Ecology Progress Series 272: 117-130 DOI 10.3354/meps272117.

7. Lesser MP, and Farrell JH. 2004. Exposure to solar radiation increases damage to both host tissues and algal symbionts of corals during thermal stress. Coral reefs 23(3): 367-377 DOI 10.1007/s00338-004-0392-z.

8. Lesser MP. 2006. Oxidative stress in marine environments: biochemistry and physiological ecology. The Annual Review of Physiology 68: 253-278 DOI 10.1146/annurev.physiol.68.040104.110001.

9. Byczkowski JZ, Gessner T. 1988. Biological role of superoxide ion radical. International Journal of Biochemistry 20: 569-580 DOI 10.1016/0020-711x (88)90095-x.

10. Runkel ED, Liu S, Baumeister R, and Schulze E. 2013. Surveillance-activated defenses block the ROS-induced mitochondrial unfolded protein response. PLoS Genet,9(3): e1003346 DOI 10.1371/journal.pgen.1003346.

11. Mayfield AB, Fan TY, and Chen CS. 2013. Physiological acclimation to elevated temperature in a reef-building coral from an upwelling environment. Coral Reefs, 32(4): 909-921 DOI 10.1007/s00338-013-1067-4.

12. Mayfield AB, Chan PH, Putnam HM, Chen CS, and Fan TY. 2012. The effects of a variable temperature regime on the physiology of the reef-building coral *Seriatopora hystrix*: results from a laboratory-based reciprocal transplant. Journal of Experimental Biology 215(23): 4183-4195 DOI 10.1242/jeb.071688.

13. Meng PJ, Lee HJ, Wang JT, Chen CC, Lin HJ, Tew KS, and Hsieh WJ. 2008. A long-term survey on anthropogenic impacts to the water quality of coral reefs, southern Taiwan. Environmental Pollution 156(1): 67-75 DOI 10.1016/j.envpol.2007.12.039.

14. Liu PJ, Meng PJ, Liu LL, Wang JT, and Leu MY. 2012. Impacts of human activities on coral reef ecosystems of southern Taiwan: a long-term study. Marine pollution bulletin 64(6): 1129-1135 DOI 10.1016/j.marpolbul.2012.03.031.

15. Chen HK, Wang LH, Chen WNU, Mayfield AB, Levy O, Lin CS, and Chen CS. 2017. Coral lipid bodies as the relay center interconnecting diel-dependent lipidomic changes in different cellular compartments. Scientific reports 7(1): 1-13 DOI 10.1038/s41598-017-02722-z.

16. Chen WNU, Kang HJ, Weis V, Mayfield AB, Jiang PL, Fang LS, and Chen CS. 2012. Diel rhythmicity of lipid-body formation in a coral-*Symbiodinium* endosymbiosis. Coral Reefs 31(2): 521-534 DOI 10.1007/s00338-011-0868-6.

## *Appendix figures for reviewer #2*

## Appendix figure 1. Purified lipid Bodies (LBs) under differential interference contrast microscopy, and different fluorescent stain.

## *Reviewer #3:*

**Basic reporting**

The report matched categories for basic reporting.

**Experimental design**

The report matched categories for experimental design.

**Validity of the findings**

I only have some concerns on the general interpretation & presentation of the data, which are included in the "general comments for the authors".

**Comments for the author**

The paper presents an interesting phenomenon, in which the formation of lipid body in coral during coral-dinoflagellate endosymbiosis, represented by LB size and distributions fluctuate by diel rhythmicity. The study also revealed critical transcriptomic regulation of lipid biosynthesis. The experiments are well-executed, with experimental procedures described in much detail in the Methods section. However, some concerns about the results need to be addressed before this manuscript can be accepted for publication in PeerJ.

1. *Line 254: Please specify the method to count the LB on mesoglea side or coelenteron side (What percentage of the LB falling into a certain side is considered counted as mesoglea or coelenteron?)*

**Author's response:**

All LBs were counted and allocated into three size classes (<1 μm, 1-3 μm, >3 μm) and two distribution categories (near mesoglea, near coelenteron). For distribution categories we defined the middle line as the midpoint between the edge of gastroderm near mesoglea to the edge near coelenteron. Image software was used to calculate the area of the LB that covered the middle line. If there was more than 50 % of the LB area within a certain side, it was accordingly counted as either mesoglea or coelenteron.

2. *Line 277-280: On Figure 4, it appears that PERK and IRE gene expression increased by one log, which means 2 folds, not one fold as stated in this sentence.*

**Author's response:**

We appreciate the reviewer's comments. This was not an accurate description of the data. **Figure 4** has been revised whereby we now include the range label of color key was changed to "-4 to 4" in **Figure 4A** and the x-axis label was changed to "fold change(log2 FPKM) to log2 FPKM" in **Figure 4B**. Also, the text has now been revised accordingly:

*Lines 283-297:* "Specifically, as a monitor of UPR activation/ER stress, the ER chaperone and signaling regulator GRP78/BiP exhibited a fourfold increase in expression at noon. Nearly simultaneously, the two sensor proteins protein kinase R-like ER kinase (PERK) and inositol-requiring enzyme 1 (IRE1), which induce various pathways and drive lipid biosynthesis, exhibited near a two-fold increase in expression. IRE1 expression started to increase at sunrise and could reach a two-fold change in transcription factor X-box binding protein 1 (XBP1) expression by noon. The eukaryotic translation initiation factor 2 subunit 1 (eIF2α) expression started to increase twofold change at sunrise. Transcription factor sterol regulatory element-binding protein 1 (SREBP1) activated by PERK also had increased expression at noon. The activating transcription factor 6 (ATF6) sensor protein was activated after midnight, whereas membrane-bound transcription factor site-1 protease (S1P) exhibited increased expression at sunset. Fig. 4B presents a box plot of the distributions and outliers of gene expression for UPR activation/ER stress genes related to LB lipogenesis. Genes including BiP, PERK, eIF2α, SREBP1, IRE1, XBP1, ATF6, and S1P were upregulated starting at sunrise and downregulated from sunset to midnight."

3. *I encourage the author to deposit the data of their RNA-Seq analysis. The supplementary data only contained a small number of genes (BIP, PERK, IRE1, etc.). A GO analysis of the RNA-Seq data could reveal even more patterns in gene expression during diel fluctuations.*

**Author's response:**

We are thankful for reviewer's comments. We have also been working on NGS-analyses of the transcriptome, including Kyoto Encyclopedia of Genes and Genomes (KEGG) pathway enrichment analysis and Gene Ontology (GO) functional

enrichment analysis; a detailed comparison of the coral host and their Symbiodiniaceae during the diel cycle will outlined in a separate paper.

4. *Figure 5A: the author should perform an analysis of colocalization between Bodipy and ER Tracker signals. This can be done with ImageJ. A merged image of these 2 channels should also be included.*

**Author's response:**

Please see our response to this concern above [see in **Author's response to "Reviewer #2 (comments to the author)"**]. According to reviewer #2 and #3's comments, we have now enhanced the images in Figure 5 and added these to "**Supplementary Figure S4.**

5. *The authors claimed micro-LBs fused to form larger LBs. I assumed that However, looking at micro-LB, can the authors provide an explanation why the number of Micro-LB did not really change much between light to dark phase. (a one-way ANOVA analysis of samples near mesoglea would be able to distinguish if there is any difference in Micro-LB number over time).*

**Author's response:**

As shown in the upper panel of **Figure 3C** "Micro-LB" and **Supplementary Table S1**, the percentage of micro-LB numbers was found to gradually increase from sunrise to midnight. We conducted a one-way ANOVA of the percentage of micro-LB near the mesoglea and found a significant difference between light and dark cycles (F=7.64; p=0.000). The average percentage of micro-LBs in the light phase, (i.e., time points: 06:00, 09:00, 12:00, and 15:00), was $40.6 \pm 2.5$ %. The percentage of micrio-LBs in the dark phase (i.e., time points: 18:30, 21:00, 00:00, 03:00, and 05:00) was $47.8 \pm 3.9$ %.

6. *Figure 2: Panels in the second and third columns appeared to be at different scale compated to the corresponding rows in the first column. For example: 2I scale bar is larger than 2G, H even though they are at the same magnification. Working with images at the inaccurate scale could lead to inaccurate*

*quantification of the LB size. I encourage the authors to double check their raw data/images to confirm this.*

**Author's response:**

We are thankful for the reviewer's comments and have checked the raw images. As you correctly pointed out, the scale bar of **Figure 2F** should be 1 µm, and **Figure 2I** was an enlargement of **Figure 2F**. The pertinent new text has now been added to the figure legend accordingly.

7.    *Figure 5B: what was the time-point chosen for this Western Blot?*

**Author's response:**

The samples for western blot were collected at six time points during the diel cycle: 06:00 (sunrise), 12:00 (noon), 15:00, 18:00 (sunset), 21:00, and 00:00 (midnight). See in lines 182-183 of "Materials & Methods" section.